# Nuclear Transporter *IPO13* Is Central to Efficient Neuronal Differentiation

**DOI:** 10.3390/cells11121904

**Published:** 2022-06-12

**Authors:** Katarzyna A. Gajewska, John M. Haynes, David A. Jans

**Affiliations:** 1Biomedicine Discovery Institute, Monash University, Clayton 3168, Australia; kasia.gajewska@monash.edu; 2Monash Institute of Pharmaceutical Sciences, Monash University, Parkville 3052, Australia; john.haynes@monash.edu

**Keywords:** nuclear transport, embryonic stem cells, neurogenesis

## Abstract

Molecular transport between the nucleus and cytoplasm of the cell is mediated by the importin superfamily of transport receptors, of which the bidirectional transporter Importin 13 (IPO13) is a unique member, with a critical role in early embryonic development through nuclear transport of key regulators, such as transcription factors Pax6, Pax3, and ARX. Here, we examined the role of IPO13 in neuronal differentiation for the first time, using a mouse embryonic stem cell (ESC) model and a monolayer-based differentiation protocol to compare IPO13^−^^/^^−^ to wild type ESCs. Although IPO13^−^^/^^−^ ESCs differentiated into neural progenitor cells, as indicated by the expression of dorsal forebrain progenitor markers, reduced expression of progenitor markers *Pax6* and *Nestin* compared to IPO13^−^^/^^−^ was evident, concomitant with reduced nuclear localisation/transcriptional function of IPO13 import cargo Pax6. Differentiation of IPO13^−^^/^^−^ cells into neurons appeared to be strongly impaired, as evidenced by altered morphology, reduced expression of key neuronal markers, and altered response to the neurotransmitter glutamate. Our findings establish that IPO13 has a key role in ESC neuronal differentiation, in part through the nuclear transport of Pax6.

## 1. Introduction

Regulated transport between the nucleus and the cytoplasm is central to the spatiotemporal regulation of proteins involved in gene expression, signal transduction, cell proliferation, and differentiation [1,2]. Molecules larger than ~40 kDa are conventionally imported into the nucleus by members of the importin (IMP) superfamily of transport proteins of which there are multiple α and β subtypes [2]. Although some redundancy in cargo recognition exists amongst members of the IMP family, many cargoes are recognised and transported by a specific IMP [3,4,5,6]. IPO13 is a unique member of the IMPβ family as it is one of only three mammalian members able to mediate nuclear transport bidirectionally and, thus, has the potential to import one cargo and export another with one nuclear transport cycle into and out of the nucleus [7,8]. IPO13 imports and exports a diverse cohort of cargoes, thereby regulating diverse cellular processes. Import cargoes of IPO13 include the SUMO E2-conjugating enzyme UBC9, the glucocorticoid receptor, and the neural development transcription factors Pax6 and ARX [9,10,11,12]. Export cargoes of IPO13 include the translation initiation factor eIF1A and EIF4G2 and the high mobility group protein 20A [7,8]. 

Stem cells are characterised by their capacity to self-renew and differentiate into distinct cell types [13]. The choreographed expression and transport into the nucleus of pluripotency or differentiation factors is a key regulatory mechanism underlying the cell fate decision [14,15]. Tight regulation of IMP localisation/expression directly impacts the localisation of specific transcription factors (TFs) and other proteins, thereby modulating TF-dependent gene expression throughout development. Neuronal differentiation of stem cells, for example, is dependent on switching the expression of importin α1 to α5, enabling the nuclear import of TFs that promote neuronal differentiation, in place of those maintaining pluripotency [16,17].

Apart from mediating the nuclear transport of many developmentally critical proteins, including the master regulator for the eye and brain development transcription factor Pax6, the expression of *IPO13* itself is developmentally regulated in foetal rat lung and foetal mouse brain developmental stages, with *IPO13* being the most expressed IPO in the neural ectoderm differentiated from mouse embryonic stem cells (mESCs), suggesting IPO13-dependent transport is critical in the developing brain [18,19,20]. That *IPO13* expression is critical during early embryonic development is indicated by the fact that IPO13 knockout is embryonic lethal. To address the contribution of IPO13 to neuronal differentiation, we utilised an IPO13 knockout (IPO13^−^^/^^−^) mESC model to examine the effects of IPO13 knockout on neuronal differentiation; the cell line has previously been used in an embryoid body model to document reduced cell proliferation concomitant with reduced mesodermal marker expression [21]. We recently used the cell line to establish IPO13’s key role in the oxidative stress response [22,23]. Here, we demonstrate for the first time that IPO13 plays a role in mESC differentiation into neurons. We found that IPO13^−^^/^^−^ mESCs can differentiate into neural progenitor cells but exhibit distinct aberrations in localisation and expression of critical neuronal factors. Likely as a consequence of this, IPO13^−^^/^^−^ cells exhibit impaired neurogenesis as evidenced by the morphological differences, decreased expression of neuronal markers, and impaired neurotransmitter response compared to IPO13^+/+^ mESCs. Our findings indicate that the IPO13-mediated nuclear transport is vital to ESC neurogenesis, consistent with the idea that IPO13 is a key player in early embryonic brain development.

## 2. Materials and Methods

### 2.1. Cell Culture and Immunofluorescence

The IPO13^+/+^ and IPO13^−^^/^^−^ mESC lines have been described previously in detail [8,21,22,23]; they were maintained in feeder-free culture conditions in DMEM supplemented with 12.5% FBS, 1 × Glutamax, 0.1 mM non-essential amino acids, 0.1 mM β mercaptoethanol, and 1000 U/mL leukaemia inhibitor factor (LIF) at 37 °C on 0.1% gelatin-coated surfaces in 5% CO_2_ atmosphere in a humidified incubator, with passaging every 2 d.

Cells were immunostained as previously [24]; briefly, cells were fixed with 4% paraformaldehyde (PFA) and permeabilised with 0.2% Triton X-100. Non-specific binding sites were blocked with blocking buffer (1% BSA in PBS) and immunostained with primary antibodies at room temperature for 1 h followed by a 1 h incubation with conjugated secondary antibodies. Cells were washed between steps with PBS. Coverslips were mounted to glass slides with ProLong Gold antifade reagent with DAPI (Invitrogen, Waltham, MA, USA). Primary antibodies specific for Oct4 (Santa Cruz, Dallas, TX, USA, sc-5279), Nestin (DSHB, Iowa City, IA, USA, rat-401), Pax6 (Thermo Fisher, Waltham, MA, USA, 42-6600), Tubb3 (Promega, G7121), and Alexa-Fluor 488-goat anti-rabbit and Alexa-Fluor 488-goat anti-mouse IgG secondary antibodies were used for immunofluorescence experiments.

### 2.2. ESC Differentiation

ESCs were seeded in 0.1% gelatinised 6-well plates at a density of 4.0 × 10^4^ cells per well and incubated overnight in a LIF-supplemented medium prior to differentiation, as previously [25]. For early patterning, serum replacement medium (SRM) and N2-supplemented medium were prepared, supplemented with 200 nM LDN193189 and 20 ng/mL fibroblast growth factor 2 (FGF2). SRM consisted of knockout DMEM, 15% knockout serum, 1 × P/S, 1 × Glutamax, 1 × NEAA, 0.1 mM β mercaptoethanol, while the N2-supplemented medium consisted of DMEM/F12, 1 × P/S, 1 × Glutamax, 1 × NEAA, 0.1 mM β mercaptoethanol, 1 × Insulin-Transferrin-Selenium-Sodium Pyruvate (ITS-A) supplement, and 1 × N2 supplement. ESCs were cultured in a progressive gradient (D0:100%SRM; D1:75%SRM:25%N2; D2:50%SRM:50%N2; D3:25%SRM:75%N2, where D represents differentiation day) of relative amounts of the two media (‘Patterning Media’). During early patterning (D0-D10); culture media (changed daily) was supplemented with LDN193189 with FGF2 added from D2. Cells were then switched to N2B27 maturation medium, consisting of a 1:1 mixture of DMEM/F12 and neurobasal medium, 1 × P/S, 1 × Glutamax, 1 × NEAA, 0.1 mM β mercaptoethanol, 1 × ITS-A, 1 × N2 supplement, and 1 × B27 + vitamin A supplement for D10 to D18 with media replaced every 2 d. N2B27 media was supplemented with glial cell-derived neurotrophic factor (GDNF, 30 ng/mL), brain-derived neurotrophic factor (BDNF, 30 ng/mL), ascorbic acid (AA, 0.2 mM), and the γ-secretase/Notch inhibitor DAPT (N-[N-(3,5-difluorophenacetyl)-l-alanyl]-S-phenylglycine t-butyl ester) (10 µM).

### 2.3. RNA Extraction and RT-PCR

Total RNA was isolated from cell lines using the Isolate II RNA Mini Kit (Bioline, London, UK) according to the manufacturer’s instructions; quantitative RT-PCR was performed as previously [22] using the C1000 Touch Thermal Cycler (Biorad, Hercules, CA, USA) for triplicate reactions to detect and compare expression levels between IPO13^+/+^ or IPO13^−^^/^^−^ ESC and D10, D14, and D18 differentiated cells. Data were normalised to the expression of the housekeeping genes *Tbp*, *Sdha*, or *18s* using the ΔΔCt method as described [26].

### 2.4. Flow Cytometry

Cells cultured for flow cytometric analysis, as previously described [27], were harvested by trypsinisation/centrifugation, and washed in PBS before fixation with 4% PFA at room temperature for 10 min. Cells were then washed in ice-cold PBS and permeabilised in 0.1% Triton-X100 for 20 min on ice. Cells were then washed in ice-cold PBS and blocked with Blocking Solution (BS, 10% FBS in PBS) for 2 h on ice. Next, the appropriate antibody was diluted in BS and added to cells. Cells were incubated for 2 h with gentle shaking. Cells were then washed with ice-cold washing solution (WS, 1% BSA, 0.025% TritonX-100) and incubated with the IgG kappa-binding protein conjugated to fluorescein (FITC) (Santa Cruz, sc-516140) in BS for 45 min. Cells were then washed with WS and resuspended in PBS. Cells positive for the antibody stain were quantified using flow cytometry (FACS Calibur, BD, Franklin Lakes, NJ, USA) and analysed using the FlowJo software (Ashland, OR, USA). Primary antibodies used in immunofluorescence experiments were used in flow cytometry experiments.

### 2.5. Live Cell Calcium Imaging

Calcium imaging was performed as previously [28]. Briefly, imaging was performed on D14 cells using a Nikon A1R confocal microscope (Minato City, Japan). Cells were loaded with 5 μM Fluo-4AM Ester for 30 min (37 °C) prior to baseline imaging (488/510–540 nm excitation/emission). Small clusters of cells were identified for the recording of fluorescence intensities. After a brief equilibration period, concentrations of glutamate or vehicle were added to each well. Following peak response, 300 mM of KCl was added. Background fluorescence was subtracted from the cluster fluorescence intensity prior to the analysis.

### 2.6. Statistical Analysis

The statistical analysis was performed using the 2-tailed Student’s *t*-test, assuming equal variances between two groups (Prism 8 Software, San Diego, CA, USA), unless otherwise stated.

## 3. Results

### 3.1. IPO13^−^^/^^−^ mESCs Can Differentiate into Neural Progenitor Cells

To explore the importance of IPO13 in neuron differentiation, we utilised the IPO13^−^^/^^−^ mESC model that has been characterised in detail, including a demonstration of negligible *IPO13* expression by western and qPCR analyses [8,21,22,23]. We examined the potential of IPO13^−^^/^^−^ mESCs to differentiate into neural progenitor cells using an established monolayer-based differentiation protocol (see schematic in Figure 1a) previously used to differentiate mESCs toward a dorsal forebrain fate [25]. IPO13^+/+^ and IPO13^−^^/^^−^ cells were cultured in patterning media for 10 days, supplemented with LDN193189 and FGF2 from day 2 onward. On differentiation day 10 (D10), IPO13^+/+^ and IPO13^−^^/^^−^ cells were collected for a flow cytometric analysis. As expected, D10 IPO13^+/+^ and IPO13^−^^/^^−^ cell cultures showed a reduced proportion of cells positive for the pluripotency marker Oct4 at D10 compared to mESCs that were not induced to differentiate (D0) (Figure 1c,e). In contrast, the neural progenitor marker Nestin could be detected in both IPO13^+/+^ and IPO13^−^^/^^−^ cells at D10 (Figure 1d,f).

These findings were confirmed by the RT-qPCR analysis of *Oct4* and *Nestin* expression levels (Figure 2a), whereby *Oct4* expression was significantly (*p* < 0.0001) reduced and *Nestin* expression was significantly (*p* < 0.0001) greater in both D10 IPO13^+/+^ and IPO13^−^^/^^−^ cells. Interestingly, *Nestin* expression was significantly higher (*p* < 0.05) in IPO13^+/+^ compared to IPO13^−^^/^^−^ D10 cells. Expression of the pluripotency genes *Nanog* and *Sox2* was also assessed in D10 cells (Figure 2a) and found to be significantly (*p* < 0.0001) lower in D10 cells compared to the D0 counterparts; *Sox2* expression was significantly (*p* < 0.05) higher in D10 IPO13^+/+^ cells compared to IPO13^−^^/^^−^ cells. We also examined the expression of the dorsal TF *Pax6*, finding that expression was significantly (*p* < 0.01) higher in D0 IPO13^−^^/^^−^ cells compared to D0 IPO13^+/+^, consistent with published sequencing data indicating upregulated expression of *Pax6* in IPO13^−^^/^^−^ mESCs relative to IPO13^+/+^ mESCs [22]. *Pax6* expression was upregulated at D10 in both IPO13^+/+^ and IPO13^−^^/^^−^ cells but to a significantly lower (*p* < 0.01) extent in the IPO13^−^^/^^−^ cells. Finally, the forebrain–midbrain marker *Otx2* was significantly (*p* < 0.05) upregulated in both cell lines at D10 compared to D0.

Consistent with the results of the qPCR analysis, immunostaining indicated reduced levels of Oct4 in D10 cells compared with D0 cells, and the presence of Nestin and Pax6 was detectable in D10 but not in D0 IPO13^+/+^ cells (Figure 2b). However, in D10 IPO13^−^^/^^−^ cells, Pax6 staining was diffused throughout the cells in contrast to D10 IPO13^+/+^ cells where Pax6 staining was more strongly associated with the nucleus. Overall, the data indicate that both IPO13^+/+^ and IPO13^−^^/^^−^ mESCs can differentiate into cells able to express markers consistent with dorsal forebrain progenitors. However, the expressions of *Sox2*, *Nestin*, and *Pax6* were significantly reduced in the IPO13^−^^/^^−^ cells, concomitant with altered, more cytoplasmic localisation of the Pax6 protein.

### 3.2. Pax6 Transcriptional Activity Is Reduced in D10 IPO13^−^^/^^−^ Cells

Since Pax6 localisation is more diffuse in D10 IPO13^−^^/^^−^ cells than in D10 IPO13^+/+^ cells (Figure 2b), we analysed images of Pax6 immunostained D10 IPO13^+/+^ and IPO13^−^^/^^−^ cells to determine the nuclear to cytoplasmic fluorescence ratio (Fn/c) for Pax6, where an Fn/c value < 1 indicated nuclear exclusion, and >1 indicated nuclear accumulation (Figure 3a,b). We found that the nuclear accumulation of Pax6 was significantly (*p* < 0.0001) reduced in IPO13^−^^/^^−^ cells compared to IPO13^+/+^ cells (Fn/c of 1.5 and 2.5; respectively), suggesting that nuclear trafficking of Pax6 was altered in this cell line, resulting in reduced nuclear accumulation of Pax6. Since IPO13 is known to be a nuclear import receptor for Pax6 [10], it seems reasonable to attribute the reduction in the nuclear localisation of Pax6 in D10 IPO13^−^^/^^−^ cells to be a direct result of the reduced IPO13-mediated nuclear import of Pax6. Consistent with an important role for IPO13 in D10 cells when Pax6 enters the nucleus, we found that the expression of *IPO13* is much greater in D10 IPO13^+/+^ cells compared to those in D0 cells (Figure 3c).

Pax6 is a key determinant of neuronal fate, determining whether progenitor cells self-renew or undergo neurogenesis [29,30,31]. As a TF, Pax6 regulates neuronal differentiation by regulating the expression of downstream effectors of neuronal differentiation. Given the decreased expression levels (Figure 2a) and reduced nuclear localisation of Pax6 in IPO13^−^^/^^−^ D10 cells (Figure 3a,b), we performed a qPCR analysis of D0 and D10 cells to assess the transcriptional consequences for direct targets of Pax6 in the mouse developing cortex and mouse embryonic stem cells, Neurog2, Kif1b, and Gad45g [29,32] (Figure 3c). TF Neurog2, a regulator of neuronal differentiation of progenitor cells [33], can generate induced neurons if overexpressed in ESCs [34]. We found that *Neurog2* expression was much greater in D10 IPO13^+/+^ cells compared to levels at D0, consistent with Neurog2’s role in neurogenesis. In stark contrast, *Neurog2* expression was not induced in D10 IPO13^−^^/^^−^ cells. Mitochondrial motor *Kif1b* [35] expression was significantly (*p* < 0.01) greater than D0 in D10 IPO13^+/+^ cells as well as D10 IPO13^−^^/^^−^ cells. Expression levels in D10 IPO13^−^^/^^−^ cells, however, were significantly (*p* < 0.05) lower than in D10 IPO13^+/+^ cells. Finally, the expression of DNA damage-inducible gene *Gadd45g* was significantly (*p* < 0.01) greater in both D10 IPO13^+/+^ and D10 IPO13^−^^/^^−^ cells, but not significantly different between the two lines. Therefore, clearly, the expression of Pax6 gene target *Neurog2* appears to be completely dependent on IPO13 in D10 cells, whereas *Kif1b* expression is partly dependent on IPO13; *Gadd45g* expression in D10 cells appears to be essentially independent of IPO13.

Overall, our findings indicate that the absence of IPO13 in the IPO13^−^^/^^−^ cells impacts Pax6 transcriptional activity, presumably the result of the reduced import of Pax6 as well as reduced *Pax6* expression. Given that Pax6 autoregulates its own expression [36,37], the absence of the IPO13-dependent import of Pax6 in D10 IPO13^−^^/^^−^ cells (Figure 3a,b) almost certainly contributes to the reduced *Pax6* expression observed in D10 IPO13^−^^/^^−^ cells (Figure 2a).

### 3.3. IPO13 Knockout Results in Reduced Neurogenic Potential of Progenitor Cells

We next examined the impact on differentiation into the neurons of D10 IPO13^−^^/^^−^ cells, which display reduced expressions of *Nestin*, *Neurog2*, and *Pax6* due to decreased transcriptional activity of Pax6. D10 IPO13^+/+^ and IPO13^−^^/^^−^ cells were induced to differentiate into neurons using a maturation medium supplemented with BDNF, GDNF, ascorbic acid, and DAPT, and subsequently analysed on differentiation day 14 (D14) and differentiation day 18 (D18) (Figure 4a). At D14, clear morphological differences were observable between IPO13^+/+^ and IPO13^−^^/^^−^ cells (Figure 4b), where IPO13^+/+^ cells appeared to form neurite networks, clustering with significant axonal projections, with complex neurite branching, contrasting the IPO13^−^^/^^−^ cells, which clumped together with markedly reduced neurite outgrowth, characterised by thinner axons. Immunostaining at D14 showed that the neuronal cytoskeletal protein/neuronal marker Tubb3 was present in both cell lines (Figure 4c), but IPO13^+/+^ cells showed much stronger staining, with thick, and uniform outward-projecting Tubb3 positive neurites; IPO13^−^^/^^−^ cell staining was much weaker. Nestin immunostaining of D14 IPO13^+/+^ and IPO13^−^^/^^−^ cells indicated that both lines had lost immunoreactivity (Figure 4c) compared to their D10 counterparts (Figure 2b). Since *Nestin* is expressed in neural progenitor cells and not in neurons [38,39], loss of Nestin in the D14 cells implies strongly that neural progenitor cells have differentiated.

Concomitant with the loss of Nestin, increased Tubb3 immunoreactivity strongly suggests that at D14, both IPO13^+/+^ and IPO13^−/−^ cells are no longer neural progenitor cells and that IPO13^+/+^ cells, and perhaps IPO13^−/−^ cells to a lesser extent, have differentiated into neurons. To confirm these findings and to further characterise D14 cells, we performed a qPCR analysis. Nestin expression was significantly (*p* < 0.0001) reduced in both IPO13^+/+^ and IPO13^−/−^ cells compared to D10 (Figure 4d). Tubb3 expression was significantly greater (*p* < 0.01) in D14 cells. However, IPO13^−/−^ cells expressed two-fold less Tubb3 at D14 (*p* < 0.05) than IPO13^+/+^ cells. Both IPO13^+/+^ and IPO13^−/−^ D14 cell cultures included maturing dorsal forebrain neurons as indicated by the expressions of mature dorsal forebrain markers Tbr1 and Ctip2 [25,40]. The GABAergic neuron markers Gad67 and Gad65 were also expressed at D14, but Gad67 levels were significantly (*p* < 0.05) (two-fold) lower in D14 IPO13^−/−^ cells compared to IPO13^+/+^ cells.

Astrocytes are generated from ESCs as part of neurogenesis [41]; to confirm that in addition to mature neurons, D10 cells could generate astrocytes, we examined the expression of astrocytic markers *Aqp4*, *S100b*, and *Glast* in D14 cells. We found that expressions of *Aqp4* and *S100b* were significantly (*p* < 0.05) greater at D14 in both cell lines. However, expressions of both *Aqp4* and *S100b* were significantly (*p* < 0.05) reduced (two-fold) in IPO13^−^^/^^−^ cells compared to IPO13^+/+^ cells. These data suggest that although both IPO13^+/+^ and IPO13^−^^/^^−^ cells may generate astrocytes, IPO13^−^^/^^−^ cells are impaired in this property. There were no significant differences in the expression levels of *Glast* in either cell line compared to the D10 counterparts. Importantly, we observed a significant (*p* < 0.05) increase (two-fold) in the expression level of *IPO13* at D14 compared to D10 in IPO13^+/+^ cells, indicating *IPO13* expression continues to increase beyond D10, presumably because there is a continuing role for the IPO13-mediated nuclear transport throughout neurogenesis.

Cells were cultured in maturation media for an additional 4 days and mRNA was extracted for further qPCR analysis (D18) (Figure 5a,b). We found that *Tubb3* expression levels continued to increase in IPO13^+/+^ cells, whereby expression levels at D18 were six-fold greater (*p* < 0.05) than at D10 compared to levels at D14 that were four-fold greater than at D10 (Figure 5b). In contrast, *Tubb3* levels in IPO13^−^^/^^−^ cells were significantly (*p* < 0.05) lower at D18. Similarly, expression of *Ctip2* and *Gad67* continued to increase in the IPO13^+/+^ cells from a 3-(D14) to 6-fold (D18) difference for *Ctip2* compared to D10 and from a 10-(D14) to 20-fold (D18) difference for *Gad67*. Expression levels of *Tbr1* and *Gad65* did not increase any further at D18 in either cell line (Figure 5b).

Astrocyte marker *Glast*, which was not significantly upregulated in D14 cells, was significantly (*p* < 0.05) upregulated in D18 IPO13^+/+^ cells, but not in D18 IPO13^−^^/^^−^ cells (Figure 5b). *S100b* expression similarly continued to increase beyond D14 in IPO13^+/+^ cells from a 6-fold increase at D14 relative to D10 to a 12-fold increase (*p* < 0.01) at D18 relative to D10. In contrast, *S100b* expression did not increase to the same extent in IPO13^−^^/^^−^ cells (Figure 5b). *Aqp4* expression in contrast was comparable at D14 and D18 in both IPO13^+/+^ and IPO13^−^^/^^−^ cells. *IPO13* expression similarly showed no marked increase in expression from D14 to D18 (Figure 4d and Figure 5b). The fact that several astrocyte markers showed reduced expressions from D14 in IPO13^−^^/^^−^ cells compared to IPO13^+/+^ cells, in a similar fashion to neuron markers, implies that—concomitant with neurogenesis—astrogliogenesis is likely to be impaired in the absence of IPO13.

Finally, to test the functionality of the neurons generated, we performed live-cell fluorescent imaging of D14 IPO13^+/+^ and IPO13^−^^/^^−^ cells by testing for response to the neurotransmitter glutamate using the fluorescent Ca^2+^ indicator dye Fluo-4 (Figure 6a–d). Clusters of D14 cells were analysed, with spontaneous elevations of [Ca^2+^] evident, whereby fluorescence could be detected in the vehicle alone sample prior to the addition of glutamate or KCl. D14 IPO13^+/+^ and IPO13^−^^/^^−^ cells were treated with increasing concentrations of glutamate followed by 300 mM KCl to establish the maximum potential of cells, with data presented as a percentage of the maximum response to KCl (Figure 6e). Concentrations of glutamate used in this experiment are within physiologically relevant extracellular levels detected in the synaptic cleft that activate glutamate receptors [42,43]. At low concentrations of glutamate, the response to glutamate in IPO13^−^^/^^−^ ESCs was not significantly different from that in IPO13^+/+^ cells. However, the magnitude of response of IPO13^+/+^ cells to 1000 μM of glutamate was significantly (*p* < 0.01) greater than that of IPO13^−^^/^^−^ cells. IPO13^+/+^ cells were not markedly responsive to 1 μM of glutamate compared to the vehicle but responded significantly (*p* < 0.05) to 10 as well as 100 μM (*p* < 0.05), and 1000 μM of glutamate (*p* < 0.05), Figure 6e, Appendix A). IPO13^−^^/^^−^ cells were not markedly sensitive to 1 or 10 μM of glutamate above the vehicle although they did respond significantly (*p* < 0.05) to 100 μM of glutamate. Taken together, these data indicate that clusters of D14 IPO13^−^^/^^−^ cells have reduced sensitivity and magnitude of response to glutamate, suggesting that the absence of IPO13 impairs the neurotransmitter response of neurons. Overall, our data suggest that IPO13 is central to efficient and functional neuronal differentiation.

## 4. Discussion

This is the first study to implicate a key role for IPO13 in neurogenesis to produce functional neurons through its role in modulating Pax6 nuclear localisation. In this study, IPO13^+/+^ mESCs could be induced to differentiate into neural progenitor cells and functional, neurotransmitter responsive neurons. In contrast, IPO13^−^^/^^−^ mESCs differentiated into progenitors with distinct aberrations in the expression and localisation of neuronal markers that generated neurons with impaired neuron marker expressions and neurotransmitter responses. That *IPO13* expression is increased in IPO13^+/+^ neural progenitor cells compared to mESCs (Figure 3c) implies that IPO13 is required to drive differentiation towards neurons with this enhanced *IPO13* expression coinciding with enhanced nuclear localisation and concomitant upregulation of Pax6; in IPO13^−^^/^^−^ progenitors, these responses were reduced (Figure 2a and Figure 3a,b). It was previously known that the regulated expression of specific Importin α subtypes is critical for cell fate decisions in ESCs to differentiate down the neuronal pathway [16,17], but this is the first example of regulation by an Importin family member.

Previous studies showed that the knockout of the key neurogenic factor Pax6 in mESCs reduces the neurogenic capacity of progenitors, with extensive cell death in neuron populations [30,44,45,46]. These findings somewhat mimic the effect of IPO13 knockout we observe here, whereby IPO13^−^^/^^−^ mESCs can generate neural progenitor cells, but the generation of neurons from these progenitors is greatly affected. That Pax6 localisation, expression, and transcriptional activity are all markedly reduced in the absence of IPO13 concomitant with reduced neurogenic capacity, as shown here, implies that a key mechanism by which IPO13 contributes to neurogenesis is through the nuclear import of Pax6 in neural progenitor cells.

This is the first study to demonstrate reduced Pax6 transcriptional activity in the absence of functional IPO13. IPO13^−^^/^^−^ progenitor cells expressed significantly less *Nestin* and *Kif1a* compared to the IPO13^+/+^ cells and failed to express the pro-neural TF, *Neurog2*, all three of which are known targets of Pax6 [29,32] (Figure 2a and Figure 3c). Expression of the neuronal markers *Ctip2* and *Tubb3* is also reduced at D14 and D18 in IPO13^−^^/^^−^ cells (Figure 4d and Figure 5b), conceivably as a direct result of the reduced levels/absence of these three proteins. It is known that, for example, Neurog2 is involved in the regulation of expression of the TF *Ctip2* [47,48]. The reduced *Tubb3* expression observed could also be explained by the effect of IPO13^−^^/^^−^ on Pax6; Tubb3-positive neurons during cortical differentiation are significantly reduced in Pax6 knockout cells [49]. Pax6 may also play a role in astrocyte differentiation through binding to promoters for several astrocyte markers, including *Aqp4* [31,50,51], whose expression is reduced in D14/D18 IPO13^−^^/^^−^ cells (Figure 4d and Figure 5b). Clearly, the absence of IPO13 impacts a key part of the transcriptional network centred around Pax6 that is essential throughout neurogenesis, highlighting the vital role IPO13 plays in neuronal differentiation, not just of neuronal cells, but also of astrocytes.

D14 IPO13^−^^/^^−^ cells exhibit a very different morphology to D14 IPO13^+/+^ cells (Figure 4b), showing thinner and fewer neurite projections (Figure 4c). The reduced *Tubb3* expression in IPO13^−^^/^^−^ cells observed here may be part of the mechanism, consistent with previous studies, where Tubb3^−^^/^^−^ cells showed a growth defect in dorsal root ganglia neurons with reduced neurite outgrowth [52]. While Kif1a is known to be critical for axon outgrowth in PVD neurons in *C. elegans* [53], it is not clear whether reduced *Kif1a* expression in IPO13^−^^/^^−^ progenitor cells contributes to the morphology observed in D14 and D18 cells.

We previously performed RNA-sequencing of IPO13^+/+^ and IPO13^−^^/^^−^ mESCs used in this study [22,23], with the findings of potential relevance to the findings here. Pathways analysis for the 874 genes expressed differentially between wild type and IPO13^−^^/^^−^ ESCs highlighted an enrichment in a number of Gene Ontology terms, including cerebellar cortex morphogenesis (GO:00216960), neuron differentiation (GO:0030182), generation of neurons (GO:0048699), neurogenesis (GO:0022008), regulation of cell projection organisation (GO:0031344), and neuron projection development (GO:0031175). A total of 43 IPO13-dependent genes were annotated to neuron projection development, which is defined as a process where the specific outcome is the formation and/or maturation of the structure of neurites. Given that a striking number of genes related to neuron projection are dependent on IPO13 for expression, the contribution of IPO13 to the morphological differences observed in D14 cells is likely to be multi-faceted with respect to the number of many critical genes that together regulate neurite outgrowth. Axon branching is central to the formation of functional neuronal networks, enabling neurons to connect to synaptic targets [54]. Thus, the observed reduced neurite outgrowth in IPO13^−^^/^^−^ cells may relate specifically to effects on these genes. IPO13-dependent genes were also enriched in pathways, including synaptic signalling (GO:0099536), chemical synaptic transmission (GO:0007268), calcium ion-regulated exocytosis of the neurotransmitter (GO:0048791), regulation of glutamatergic synaptic transmission (GO:0051966), and regulation of synapse structure or activity (GO:0050803). Thus, the altered response of the IPO13^−^^/^^−^ cells to glutamate (Figure 6e) may be explained by the fact that a number of genes related to neurotransmission are impacted by the absence of IPO13 in the knockout cells. Notably, there is some evidence that suggests IPO13 is involved in synaptic transmission in the Drosophila larval neuromuscular junction, whereby IPO13 regulates presynaptic neurotransmitter release [55]. Strikingly, of the 874 IPO13-dependent genes, 133 and 141, respectively, can be identified as targets of the Pax6 and Neurog2 TFs, as determined by chromatin immunoprecipitation (ChIP) [22,33,56]; strikingly, 41 of these are targets of both TF. The clear implication here is that the IPO13-dependent nuclear import of Pax6 potentiates the expression of many genes, including the Pax6 gene target *Neurog2*, in turn impacting its gene targets during neurogenesis. A detailed examination of the combined contribution of Pax6 and Neurog2 in IPO13-dependent transcriptional networks in neurogenesis is the focus subject of further study in this lab.

Our findings together indicate that the gene regulatory network that promotes neuronal differentiation of embryonic stem cells is supported by the IPO13-dependent nuclear transport, which includes the nuclear import of Pax6 in neural progenitor cells.

## Figures and Tables

**Figure 1 cells-11-01904-f001:**
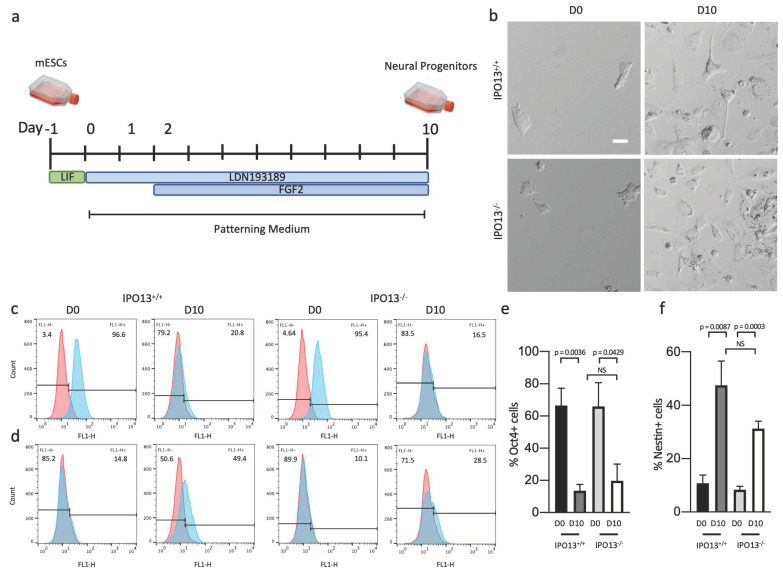
IPO13^+/+^ and IPO13^−^^/^^−^ mESCs differentiate into neural progenitor cells by D10. IPO13^+/+^ and IPO13^−^^/^^−^ mESCs were induced to differentiate toward a dorsal forebrain fate using a monolayer-based differentiation protocol and patterning medium supplemented with LDN193189 and FGF2 (**a**,**b**). Representative images of IPO13^+/+^ and IPO13^−^^/^^−^ mESCs imaged live at D0 and D10 post differentiations using phase contrast. (**c**–**f**). Flow cytometry analysis of Oct4 and Nestin in IPO13^+/+^ and IPO13^−^^/^^−^ mESCs at differentiation D0 and D10. (**c**,**d**). Representative histograms of Oct4 (**c**) and Nestin (**d**) stained cell populations (blue) and FITC alone control (red). (**e**,**f**). Pooled data mean ± SEM (*n* = 4) for % Oct4 (**e**) or Nestin (**f**) positive cells from analysis, such as that in (**c**,**d**). *p* values (2-tailed Student’s *t*-test) are indicated. Scale bar (10 μm).

**Figure 2 cells-11-01904-f002:**
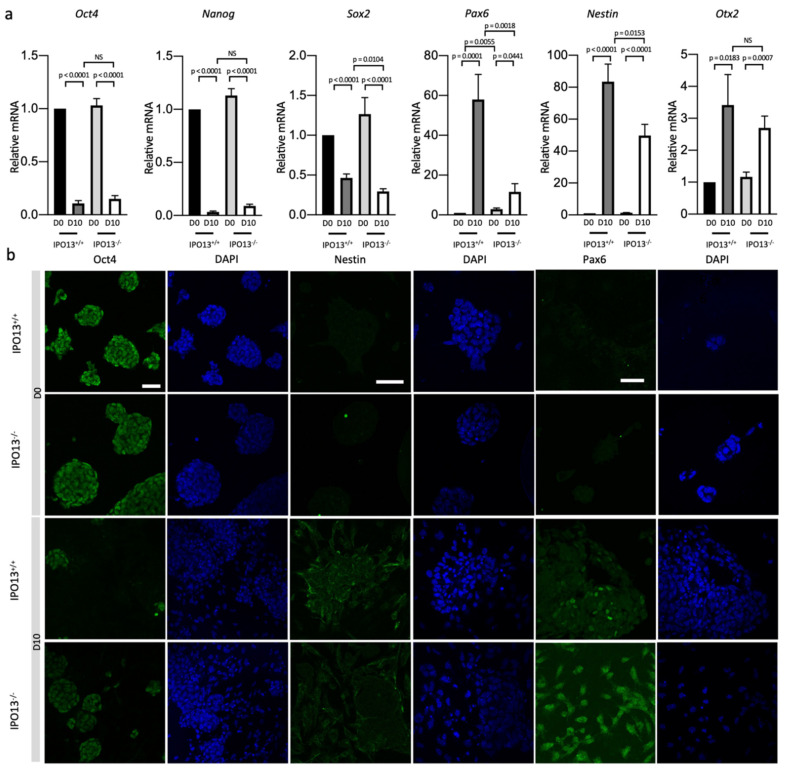
Expressions of neural progenitor markers Pax6 and Nestin were reduced in differentiated IPO13^−^^/^^−^ cells. IPO13^+/+^ and IPO13^−^^/^^−^ mESCs were induced to differentiate as in Figure 1. (**a**) RNA was extracted from IPO13^+/+^ and IPO13^−^^/^^−^ mESC at differentiations D0 and D10. Transcript levels were assessed using SensiMix SYBR Green. Results for gene expression were normalised to the expression of housekeeping genes *Sdha*, *Tbp*, and *18s*. Data were relativised to IPO13^+/+^ D0 expression values. Results represent the mean ± SEM (*n* = 7, where each cDNA sample was analysed in duplicate). *p* values as per Figure 1d. (**b**). Representative images of Oct4, Nestin, and Pax6 immunostaining from differentiation D0 and D10 in IPO13^+/+^ and IPO13^−^^/^^−^ cells. Scale bars (10 μm) are displayed on the top left image and applied to every two columns.

**Figure 3 cells-11-01904-f003:**
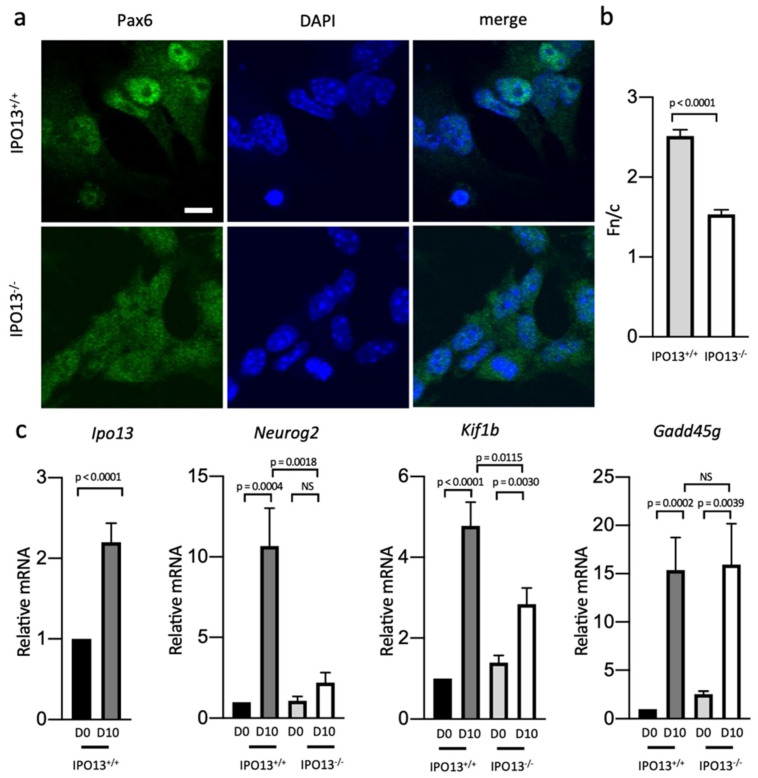
Decreased Pax6 nuclear accumulation in D10 IPO13^−^^/^^−^ cells results in reduced Pax6 transcriptional activity. (**a**). Representative images of Pax6 immunostaining at D10 in IPO13^+/+^ and IPO13^−^^/^^−^ cells. Scale bar = 10 μm. (**b**). Quantitative analysis of Pax6 localisation was carried out using the ImageJ software on images, such as those in (**a**) to determine the nuclear to cytoplasmic ratio (Fn/c). Data represent the mean ± SEM for the nuclear (Fn) and cytoplasmic fluorescence (Fc) above background fluorescence. *p* values as per Figure 1d (**c**). RNA was extracted from IPO13^+/+^ and IPO13^−^^/^^−^ mESC at D0 and D10. Transcript levels were assessed using the SensiMix SYBR Green. Results for gene expression were normalised to the expression of housekeeping genes *Sdha*, *Tbp*, and *18s*. Data were relativised to the IPO13^+/+^ D0 expression values. Results represent the mean ± SEM (*n* = 7, where each cDNA sample was measured in duplicate). *p* values are as per Figure 1d.

**Figure 4 cells-11-01904-f004:**
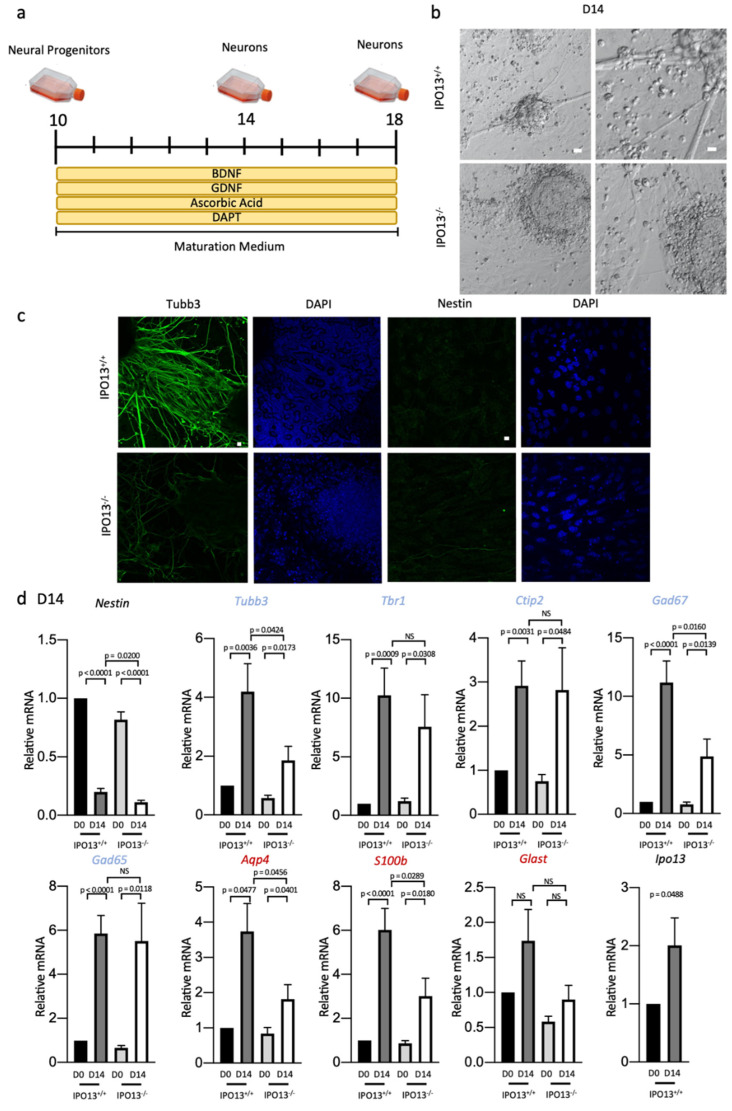
Absence of IPO13 impairs differentiation of progenitor cells to neurons at D14. IPO13^+/+^ and IPO13^−^^/^^−^ D10 cells were induced to differentiate toward the dorsal forebrain neuron fate (**a**). (**b**). Representative images of IPO13^+/+^ and IPO13^−^^/^^−^ cells imaged at differentiation D14 using phase contrast. Scale bar = 50 μm. (**c**). Representative images of Nestin and Tubb3 immunostaining from D14 in IPO13^+/+^ and IPO13^−^^/^^−^ cells. Scale bar = 10 μm. (**d**). RNA was extracted from IPO13^+/+^ and IPO13^−^^/^^−^ cells at differentiation D10 and D14. Transcript levels were assessed using SensiMix SYBR Green. Gene names in blue and red denote neuronal and astrocyte markers, respectively. Results for gene expression were normalised to the expression of housekeeping genes *Sdha*, *Tbp*, and *18s*. Data were relativised to IPO13^+/+^ D10 expression values. Results represent the mean ± SEM (*n* = 5, where cDNA samples were measured in duplicate). *p* values as per Figure 1d.

**Figure 5 cells-11-01904-f005:**
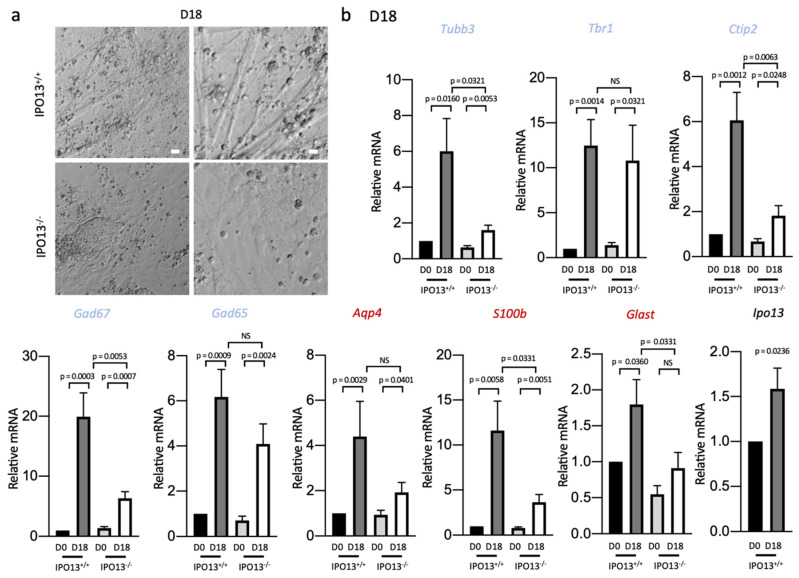
Absence of IPO13 impairs differentiation of progenitor cells to neurons at D18. IPO13^+/+^ and IPO13^−^^/^^−^ D10 cells were induced to differentiate toward the dorsal forebrain neuron fate (see Figure 4a). (**a**). Representative phase-contrast images of IPO13^+/+^ and IPO13^−^^/^^−^ cells imaged at differentiation D18 using phase contrast. Scale bar = 50 μm (**b**). RNA was extracted from IPO13^+/+^ and IPO13^−^^/^^−^ cells at D10 and D18. Transcript levels were assessed as in Figure 4d and normalised to the expression of housekeeping genes *Sdha*, *Tbp*, and *18s*. Gene names in blue and red denote neuron and astrocyte markers, respectively. Data were relativised to IPO13^+/+^ D10 expression values. Results represent the mean ± SEM (*n* = 4, where each cDNA sample was measured in duplicate). *p* values as per Figure 1d.

**Figure 6 cells-11-01904-f006:**
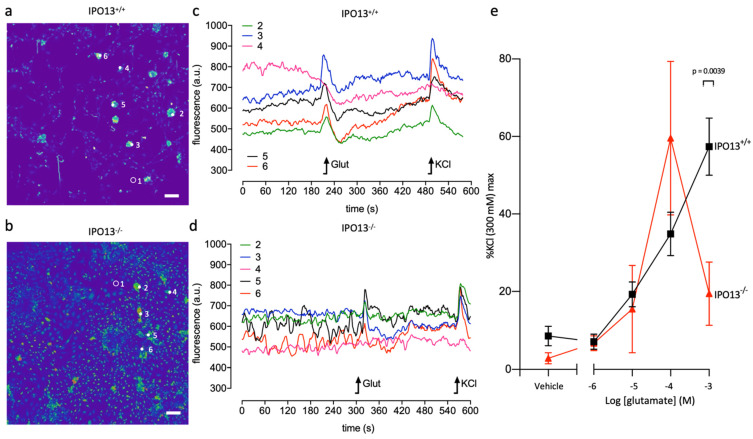
Absence of IPO13 impairs glutamate response in D14 cells. Ca^2+^ imaging was employed to examine the functions of D14 IPO13^+/+^ and IPO13^−^^/^^−^ cells. Cells were loaded with the Ca^2+^-sensitive probe Fluo-4 followed by stimulation with increasing concentrations of glutamate. (**a**,**b**). A typical field of view of IPO13^+/+^ (**a**) or IPO13^−^^/^^−^ (**b**) D14 cells stimulated with 1000 μM of glutamate. Scale bar = 100 μm. Representative clusters measured for response to glutamate are circled and numbered (1 indicates a region for background subtraction). (**c**,**d**). Representative traces from the Ca^2^ imaging experiment are shown for IPO13^+/+^ (**c**) and IPO13^−^^/^^−^ (**d**) with oscillations resulting from glutamate and KCl indicated by a black arrow. Traces are numbered, corresponding to cell clusters in (**a**,**b**,**e**). Pooled data represent the mean ± SEM of three individual experiments (*n* = 3) across which ≥ 5 randomly selected cell clusters (per n) were analysed. Statistical differences were determined by the unpaired *t*-test with Welch’s correction.

## Data Availability

Not applicable.

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
