# Peer review of "Nuclear Transporter IPO13 Is Central to Efficient Neuronal Differentiation"

_cells, 2022, doi:10.3390/cells11121904_

Round 1
Reviewer 1 Report
In this study authors investigate the role by nuclear transport protein IPO13 towards a neuronal two step differentiation protocol in ESCs derived from WT or IPO13-/- cells, respectively. Expression of dorsal forebrain progenitor markers Pax6 and Nestin was decreased compared to IPO13+/+ cells indicating that differentiation into neuronal progenitor cells was reduced in IPO13-/- ESCs. In addition, also maturation of neural progenitors into neurons was strongly impaired in IPO13-/- cells, as shown by altered morphology, reduced expression of key neuronal markers and altered response to the neurotransmitter glutamate.
A major point is concerning Figures 2 and 3: In my file, the data and the images corresponding to the legends of figure 2 are provided. However, in figure 3 the data for decreased pax6 nuclear accumulation in D10 IPO13-/- cells resulting in reduced pax6 transcriptional activity are missing. Obviously, data and images provided for figure 3 are identical to those shown already in figure 2. Thus, I am unable to evaluate data to figure 3, neither the text associated to figure 3 nor the interpretation concerning pax6 nuclear accumulation. Unfortunately, reduced pax6 transcriptional activity in D10 IPO13-/- cells may turn out to be a crucial point for the conclusions of this manuscript.
Minor points:
Figure 1: In figure 1d diagrams of flow cytometric analysis of Nestin stained cells are shown. In the first three diagrams (from left to right) for D0 and D10 of IPO13+/+ cells and for D0 in IPO13-/- cells, respectively I can see two populations each stained in red and blue indicating FITC versus Nestin stained cells. Overlapping regions between red and blue populations appear in dark blue, accordingly. In the last diagram 1d for D10 IPO13-/- cells I can identify a third peak flanking at the right side. Please explain this additional peak in the context of an increased number of Nestin + cells claiming that their differentiation is reduced.
Figure 6: You show data using 1mM glutamate to stimulate cells for calcium imaging. Can you demonstrate the specificity of the ligand to glutamate receptor interaction as the concentration of glutamate is very high? Which type of glutamate receptor do you activate?
Author Response
We thank the Reviewer for positive comments. We respond below to the points raised:
- Absence of Figure 3 ..... We understand that the Reviewer may have been provided with a version of the manuscript where Figure 2 had been mistakenly inserted amongst the text twice in the place of Figure 3 - we thought we had successfully amended this before the paper went out for review, but perhaps the Reviewer was sent the older uncorrected version of the manuscript. We apologise for any confusion and inconvenience the Reviewer experienced; we are sure Figure 3 supports all of our conclusions.
- Nestin expression in KO cells ....
We document the low level of Nestin expression in the KO cells not only in Fig. 1d, but definitively in Figure 2a, as we discuss in the results section immediately following the figure. Our data further shows that IPO13-/- cells also express lower levels of the progenitor markers Pax6 and Otx2 (Figure 2), and clear defects in both nuclear localisation and transcriptional activity of Pax6 (Figure 3 – apologies again for its absence in the original submission !). Thus, we believe that the IPO13 KO affects the neurogenic potential of the D10 cells, rather than reducing the differentiation of mESCs to progenitors. We thank the Reviewer for the opportunity to clarify this.
- Glutamate concentrations .... The concentrations of glutamate used in calcium imaging in Figure 6 are within physiologically relevant levels that activate glutamate receptors. We point this out in text relating to. the figure, and include references to support this (Lines 366-367). We also reference studies (References 41 and 42) that utilize these concentrations of glutamate, including > 1 mM glutamate to which ionotropic transmembrane receptors AMPA are responsive. We thank the Reviewer again for the constructive comments (and apologies for Figure 3 again. !!).
Reviewer 2 Report
Manuscript entitled: “Nuclear transporter IPO13 is central to efficient neuronal differentiation” by Gajewska K.A. et all, described a role of bidirectional transporter importin13, unique member of the importin B transport receptors family, in early embryonic development by regulation nuclear transport of transcription factor Pax6.
Authors of the work uncovered a key role of IPO13 in the gene regulatory network, in nuclear transport of Pax6 by showing that decreased accumulation of the transcription factor reduces its transcriptional activity in absence of the importin what is reflecting in attenuate differentiation process of neural progenitor cells to neurons as well as it impairs glutamate response of the cells.
In my opinion the studies are well designed, carefully performed, the data are solid, and the conclusions drawn are firmly based on the data obtained.
However, I have noticed a lack of any point in discussion about the exporting activity of IPO13 in this work. I feel as if it was ignored. KO IPO13 blocks not only nuclear import of IPO13 but also its export activity towards e.g. the translation initiation factor eIF1A. Are there any consequences in the promotion of neural differentiation in the absence of this activity? I am also curious to answer this question?
Author Response
We thank the Reviewer for the positive comments.
We thank the Reviewer for pointing out that the export activity of IPO13 was not sufficiently discussed. We have amended this oversight, including additional information and references for IPO13’s import and export role (Lines 33-39). We agree with the Reviewer that aside from Pax6, knock-out of IPO13 likely affects the localisation and activity of a vast number of IPO13 cargoes during neuronal differentiation, including EIF1A, which is the topic of further investigation in this lab.
We thank the Reviewer.
Round 2
Reviewer 1 Report
The paper covers an interesting topic and the experimental techniques applied are up to date. One minor point on nestin expression in Fig 1d was not really answered but the overall evidence for its down regulation was clearly given.